# The Pathways from Type A Personality to Physical and Mental Health Amid COVID-19: A Multiple-Group Path Model of Frontline Anti-Epidemic Medical Staff and Ordinary People

**DOI:** 10.3390/ijerph18041874

**Published:** 2021-02-15

**Authors:** Peng Wang, Rong Wang, Mei Tian, Yu Sun, Jiexin Ma, Yitong Tu, Yun Yan

**Affiliations:** 1School of Psychology, Shandong Normal University, Jinan 250358, Shandong, China; pengsdnu@163.com (P.W.); Wangrong_0614@163.com (R.W.); Universunny@163.com (Y.S.); MJXsdnu@163.com (J.M.); TYTsdnu@163.com (Y.T.); y15092146080@163.com (Y.Y.); 2Library, Shandong Normal University, Jinan 250358, Shandong, China

**Keywords:** COVID-19, Type A personality, frontline anti-epidemic medical staff, depression, appetite and sleep disorder, AST

## Abstract

During the COVID-19 pandemic, a survey was conducted using the questionnaire method among participants consisting of both ordinary people (*n* = 325) and frontline anti-epidemic medical staff (*n* = 310), and physiological data was obtained on the basis of physical examination. This study aimed to scrutinize the influence of Type A personality on the biochemical indicators of aspartate aminotransferase (AST) and the behavioral indicators of appetite and sleep disorder, and to analyze the mediating effect of depression. Meanwhile, multiple-group path analysis was used to evaluate path differences between the models of two samples. The results of the mediation analysis for both samples demonstrated that depression significantly mediated the relationship between Type A personality and appetite and sleep disorder. The results of multiple-group path analysis showed that the relationship between Type A personality and appetite and sleep disorder seems to be significantly stronger in ordinary people, whereas the relationship between depression and appetite and sleep disorder, as well as with the path towards AST, seems to be significantly stronger in frontline anti-epidemic medical staff. This paper provides ideas for the selection and distribution of medical personnel based on personality characteristics in major public health emergencies, and physical and mental health status should be taken into account to provide relative health assistance.

## 1. Introduction

The Coronavirus Disease 2019 (COVID-19) was declared a pandemic by the World Health Organization (WHO) on 11 March 2020, indicating that there were over 118,000 cases in over 110 countries and territories all over the world at that time [1]. Governments around the world are implementing various containment measures, and the healthcare system is preparing itself for a tsunami of people seeking treatment.

Since the outbreak of the COVID-19, global mental health has increasingly been affected by information and concerns such as stress, anxiety, depressive symptoms, insomnia, denial, anger [2]. In particular, health workers from all walks of life have been facing great challenges. Studies have shown that health professionals are at great risk in terms of their physical and mental health due to the specific stressors they are exposed to over time [3]. Meanwhile, research by Huang et al. (2020) showed that healthcare workers were at high risk for poor sleep quality and mental illness compared with other occupational groups [4]. The physical and mental health of medical personnel in the context of COVID-19 has drawn increasing attention from society. Therefore, this study focuses on front-line anti-epidemic medical personnel and ordinary people.

Friedman et al. (2014) illustrated that one way of improving health and reducing the risk of death is to study the relationship between personality and health; moreover, personality and health modeling can help clarify causal relationships and provide a possible basis for successful intervention [5]. Previous studies have explored the relationship between the Big Five personality [6,7,8], the Eysenck personality [9,10,11] and health behavior and risk of death. At the same time, individuals with Type A personality behavior characteristics face stressful events with greater work pressure and lower mental health level [12]. Furthermore, the behavioral patterns of Type A personality traits (study of which exploded in the 1950s [13] was proven in the 20th century to predict health problems such as cardiovascular disease, coronary heart disease and other heart diseases [14]. Presumably, this means that Type A personality may predict people’s physical and mental health under the stress of COVID-19. Therefore, we took Type A personality as an independent variable to explore its mechanism of explaining physical and mental health.

The large number of depressive symptoms occurring during the current global COVID-19 outbreak is also a very relevant part of the study. Mental health problems can indirectly or directly affect any body system, such as the neuroendocrine system, immune and inflammatory system, metabolic system, cardiovascular and respiratory system, and nervous system [15]. Booth-Kewley et al. (1987) showed that depressive symptoms are predictors of heart disease and cardiovascular disease [16], and this has been widely confirmed [17,18,19], causing a wave of prevention through treatment of depression. Friedman et al. (2014) pointed out that the simple (depression → disease) model is wrong, or at least incomplete, and that the relationship between depression and death is confounded by factors such as personality and unhealthy behaviors [5].

Friedman et al. (2014) proposed a broader and more comprehensive causal model between personality and various mediating and moderating variables, in which personality traits were taken as independent variables [5]. In addition, previous studies have used negative psychological reactions (i.e., depression) as mediating variables for model analysis [20,21]. Therefore, in this study, Type A personality was taken as the independent variable and depression was a mediating variable.

Aspartate aminotransferase (AST) mainly exists in the mitochondria and cytoplasm of stem cells, mostly mitochondria, and is an indispensable enzyme in the process of hepatocyte egg-white synthesis [22]. Moreover, studies have shown that abnormal AST represents the risk of fatty liver disease [23]. Ahmadi et al. (2019) pointed out that after controlling for the influence of confounding factors, stress was significantly positively correlated with the biochemical factor AST [24], and the biochemical indicator AST may be a new target for depression [25]. Therefore, based on the stress background of the COVID-19 pandemic, this study investigates the mechanism of action between Type A personality, depression and biochemical indicator AST.

At the same time, based on previous studies on the relationship between the quality of sleep state and the occurrence and maintenance of Post-Traumatic Stress Disorder (PTSD) [26], it was found that during COVID-19, participants with poor sleep state (suffering from sleep disorders and other disease) had a higher prevalence of post-traumatic stress symptoms [27]. Based on the stress background of Coronavirus Disease 2019, this study uses the biochemical factor indicator AST as the dependent variable of physical health, and the behavioral indicators appetite and sleep disorder as the dependent variables of behavioral health, in order to explore the influencing mechanism of Type A personality and depression.

Accordingly, the following hypotheses are proposed in this study: this study proposes to construct a model with depression as mediator, Type A personality as independent variable, and biochemical indicator AST and behavioral indicator appetite and sleep disorder as dependent variables. We expect that depression plays a mediating role between Type A personality and biochemical indicator AST, appetite and sleep disorder. Moreover, we also examined the differences in each path of the model between two samples (the sample of front-line anti-epidemic medical personnel and the sample of ordinary people). We expect that there are significant differences in each path of the model between the two samples.

In sum, this study proposes to explore the internal mechanism of physical and mental health changes of individuals with Type A personality traits, so as to evaluate path differences between the models of two samples by using multiple-group path analysis.

## 2. Materials and Methods

### 2.1. Participants

From 16 April 2020 to 28 June 2020, a total of 1245 participants, including frontline anti-epidemic medical staff working in an affiliated hospital and the Centers for Disease Control and Prevention as well as ordinary people, were assessed with respect to physical and mental health indicators. These answers were collected in person; specifically, the participants answered the questions using a computer, and the physiological indicators were collected one on one. According to the questionnaire response rate (i.e., whether all questions were completed), and the filtering and matching of participants, the final sample consisted of frontline anti-pandemic medical staffs (*n* = 310) and ordinary people (*n* = 325). All participants, 225 males and 410 females (*M*_age_ = 33.75; *SD* = 7.750), were assessed with respect to AST, Type A personality, depression and appetite and sleep disorder.

### 2.2. Measures

#### 2.2.1. Type A Behavior Pattern Scale

The Type A Behavior Pattern Scale [28] was developed by the National Psychosomatic Medicine Collaboration group that is comprised by 60 questions in total, including three scales. The TH (Time hurry) scale features aspects such as time urgency and doing things quickly; the CH scale features aspects such as competitiveness, hostility and lack of features such as patience; and the L (lie) scale is a determination of cover degree, with results of L ≥ 7 being invalid. Furthermore, the total score for this measurement tool was positively correlated with Type A personality: A1 (50~36 points), A2 (partial Type A personalities; 35~28 points), X (middle type personality; 27 points), B1 (partial Type B personality; 26~19 points), B2 (Type B personality; 18~1 points). The Cronbach’s coefficient of the total score of Type A behavior was 0.843. Furthermore, by comparing the test results of respondents from the southern and northern regions of China, the overall results showed that northerners had significantly higher results than the southerners, indicating that the scale has good discriminative validity [29].

#### 2.2.2. Self-Rating Depression Scale (SDS)

For assessment of participants’ subjective view of their depressive symptoms, we used the SDS [30]. The SDS contains 20 items and its design is based on the diagnostic criteria for depression. Each item of the Zung Self-Rating Scale for depression is related to one symptom of depression, which can be categorized as affective symptoms, somatic symptoms, psychomotor neurological disorders and psychological disorders. Participants rate each item with regard to how they felt during the past several days using a 4-point Likert scale. The original sum score of the SDS ranges from 20 to 80, but the result is expressed as the SDS Index, which is obtained by converting the original score into a 100-point scale. The Cronbach’s coefficient of the total score of SDS was 0.800. Moreover, according to the depression index, the respondents were divided into a depressed group and a non-depressed group for discriminant validity testing. The results showed that the scale had good discriminant validity [31].

#### 2.2.3. Appetite and Sleep Disorder

The Clinical Symptoms Self-Rating Scale (SCL-90) [32], a 90-item self-report symptom inventory, was used to assess psychological distress and symptoms of psychopathology. Participants rate each item with regard to how they felt during the past several days using a 5-point Likert scale. This scale produces ten symptom dimensions. Seven items assess disturbances in appetite and sleep disorder (Cronbach’s α = 0.763). Meanwhile, the correlation between the subscale and the total score was 0.82, indicating that the structural validity of the measurement of appetite and sleep disorder in SCL-90 was also good [33].

#### 2.2.4. AST Detection Method

A Beckman AU-2700 automatic biochemical analyzer was used to detect AST in fasting elbow venous blood, and SPSS 21.0 was used for data processing according to the standards formulated by Guan (2013). Specifically, the AST ranges (15~40, 13~35) of normal biochemical indicators for males and females, respectively, were classified and coded [34], so that continuous variables were changed into dichotomous variables (0/1) on the basis of normal/abnormal.

### 2.3. Statistical Analysis

Psychological professionals adopt unified instructions when conducting group measurements on subjects as an individual unit. All participants completed the questionnaire and AST measurement independently. The SPSS 21.0 statistical analysis software package was used for data processing, and the Mplus 7.0 structural equation software package was used for path analysis.

Initially, an independent samples *t*-test was conducted to calculate the differences in the variables between the frontline anti-epidemic medical staff and the ordinary people. Then, analyses were carried out to determine the correlation among the variables. Subsequently, mediation models were created in order to achieve the purpose of this study, and multi-group path analysis was used to check whether the structure weight was constant between different groups.

It is worth noting that, during the course of analysis, since the independent variables and mediating variables were continuous variables, but the dependent variable contained classified variables and continuous variables, the analyses were carried out according to the Mplus statements of mediation analysis, with the dependent variable as a classified variable, as described by Liu et al. (2013) (see Appendix A) [35]. In addition, Wang (2011) pointed out that when using ML estimation to analyze categorical outcome measures in Mplus, commonly used model fitting indexes and model correction indexes are not provided in the output results [36]. Geiser (2013) confirmed this statement and further pointed out that the absolute fitting assessment parameters of the model could be obtained indirectly through Mplus Monte Carlo simulation study [37]. Therefore, the absolute fitting assessment parameters of the model were analyzed according to the Monte Carlo simulation statement for the mediation fitting analysis given by Thoemmes et al. (2010) (see Appendix B) [38].

## 3. Results

### 3.1. Path Analysis for the Ordinary People and the Frontline Anti-Epidemic Medical Personnel Samples

In the results of the independent samples *t*-test, there was a significant difference between ordinary people and frontline anti-epidemic medical personnel sample in terms of Type A personality (*t* = 3.768, *p* < 0.001, Cohen’s d = 0.299), depression (*t* = 3.957, *p* < 0.001, Cohen’s d = 0.314), AST (*t* = 5.517, *p* < 0.001, Cohen’s d = 0.211). The Pearson’s correlation coefficients for each sample are presented in Table 1.

The correlation analysis results show that in both the ordinary people and the medical personnel samples, Type A personality was significantly positively correlated with depression and appetite and sleep disorder; moreover, depression was significantly positively correlated with appetite and sleep disorder.

A single-group path model was tested for the samples of ordinary people and frontline anti-epidemic medical personnel separately. The model of each sample, and the baseline model in which the two path models were combined into one model, comprise a saturated model.

Firstly, for the ordinary people sample, Type A personality was significantly positively associated with depression (β = 0.493, *p* < 0.001) and appetite and sleep disorder (β = 0.278, *p* < 0.001). At the same time, depression was significantly and positively associated with appetite and sleep disorder (β = 0.013, *p* < 0.001) (Figure 1).

Secondly, in the frontline anti-epidemic medical personnel sample, Type A personality was significantly and positively associated with AST (β = 0.182, *p* = 0.045), appetite and sleep disorder (β = 0.152, *p* = 0.002), and depression (β = 0.199, *p* = 0.001). Meanwhile, depression was significantly and positively associated with appetite and sleep disorder (β = 0.516, *p* < 0.001), and AST (β = 0.318, *p* = 0.016) (see Figure 2).

### 3.2. Mediating Analysis

To test the mediating effects in the pathway from Type A personality to AST and appetite and sleep disorder, 95% bias-corrected confidence intervals (BC CI) were calculated for both groups, using a bootstrapping method with 1000 re-samples. The estimates, standard errors, and CIs of the mediation models are presented in Table 2.

Depression significantly mediated the relationship between Type A personality and appetite and sleep disorder in the ordinary people sample and the frontline anti-epidemic medical personnel sample. There was no significant indirect effect of Type A personality on AST through depression (see Table 2).

### 3.3. Multiple-Group Path Analysis

Multiple-Group Path Analysis was used to test whether there were differences in the final model between the ordinary people and the frontline anti-epidemic medical personnel groups. This was used to study whether the two samples were similar in terms of the relationship between Type A personality, AST and appetite and sleep disorder.

Wang (2013) pointed out that when limiting the cross-group invariance of the same path coefficients of multiple groups of SEM models, other coefficients still remain different across the groups when testing the cross-group invariance of path coefficients. The direct effects of the restricted SEM model and the unrestricted SEM model were determined by investigating the changes in the models’ χ^2^ statistic, and the comparison of indirect effects was performed using Wald χ^2^ test [36].

The comparison results of direct effects showed that the parameters coefficient in the path between Type A personality and appetite and sleep disorder (χ^2^ = 9.689, *p* = 0.021), depression and appetite and sleep disorder (χ^2^ = 11.413, *p* < 0.001), depression and appetite and sleep disorder (χ^2^ = 4.087, *p* = 0.043) had statistically significant differences between the two samples (see Table 3).

Specifically, this suggests that the relationship between Type A personality and appetite and sleep disorder seems to be significantly stronger in ordinary people (β = 0.278, *p* < 0.001) than frontline anti-epidemic medical personnel (β = 0.152, *p* < 0.001). The relationship between depression and appetite and sleep disorder, AST seems to be significantly stronger in frontline anti-epidemic medical personnel (β = 0.516, *p* < 0.001; β = 0.318, *p* = 0.016) than ordinary people (β = 0.258, *p* < 0.001; β = −0.079, *p* = 0.445). The other path coefficients were found to be invariant across the samples.

Meanwhile, for the indirect effect of path model, differences in indirect effects did not show statistical significance, thus manifesting in Type A personality → depression→ AST (Waldχ^2^ = 3.115, *df* = 1, *p* = 0.0776), Type A personality → depression → appetite and sleep disorder (Waldχ^2^ = 1.058, *df* = 1, *p* = 0.304).

## 4. Discussion

### 4.1. Mechanism of Type A Personality in the Context of the COVID-19 Pandemic

The COVID-19 pandemic is having a profound impact on all aspects of society, including mental and physical health. The participants in this study are front-line anti-epidemic medical staff in a hospital in China, as well as ordinary people. Through correlation analyses, the relationships among Type A personality, biochemical indicators AST, behavioral indicators appetite and sleep disorder, and depression were verified. Specifically, in the sample of ordinary people, Type A personality significantly positively predicted depression and appetite and sleep disorder. At the same time, the findings indicated the predictive effect of depression on appetite and sleep disorder. In the frontline anti-epidemic medical staff sample, Type A personality was able to significantly predict AST, appetite and sleep disorder, and depression. Meanwhile, depression was significantly positively predicted appetite and sleep disorder and AST.

In the context of the COVID-19 pandemic, the research results of Type A personality significantly influence mental health status, which is consistent with previous studies [39] suggesting that Type A personality can predict changes in mental health status. At the same time, this study shows that Type A personality can predict the state of appetite and sleep disorder, which is a behavioral health indicator. These results were confirmed in two samples (sample of frontline anti-epidemic medical personnel and sample of ordinary people), indicating that attention should be paid to the mental health status and appetite and sleep disorder of Type A personality groups during the COVID-19 pandemic.

Previous studies on the relationship between Type A personality and biochemical indicators have mostly focused on its prediction of cardiovascular disease, coronary heart disease and other health problems [14]; this study focuses on biochemical factors associated with stress events, AST [24], to reflect the impact of Type A personalities on physical fitness. According to the results in frontline anti-epidemic medical personnel, Type A personality significantly predicts the change in biochemical index AST. However, the result was not found in the sample of ordinary people. Therefore, additional attention needs to be paid to the physical health of medical personnel with Type A personality traits during periods of major public health emergencies.

### 4.2. Path Analysis and Mediating Effect Analysis of Depression

The results of cross-sectional survey in this study showed that depression significantly positively predicted appetite and sleep disorder in both samples (the frontline anti-epidemic medical personnel sample and that of ordinary people). These results suggest that changes in depression level also have an impact on appetite and sleep disorder [40,41]. Therefore, in the case of major public health emergencies, it is important to pay attention to appetite and sleep disorder for those with higher levels of depression, and provide them with timely mental health assistance.

However, Ting et al. (2020) proposed that AST may be a new target for depression [25], which was only confirmed in the samples of frontline anti-epidemic medical workers, in which depression significantly positively predicted the abnormality of biochemical indicator AST. This indicates that, in the case of major public health emergencies, it is more important to provide mental health assistance to frontline anti-epidemic medical personnel, because its impact on such personnel is not only reflected in changes appetite and sleep disorder, but is also reflected in their physical health indicators. Frontline anti-epidemic medical workers are the main groups that require psychological crisis intervention during major public health emergencies.

This study verified through cross-sectional survey that depression plays a significant mediating role in the relationship between Type A personality and the biochemical indicator AST and the behavioral indicator appetite and sleep disorder. This study is consistent with the results of previous studies, suggesting that mental health problems can indirectly or directly predict individual behaviors [27]. Specifically, between the two samples (the sample of frontline anti-epidemic medical personnel and the sample of ordinary people), depression played a significant mediating role between Type A personality and the behavioral indicators of appetite and sleep disorder. This indicates that Type A personality can significantly positively predict changes in appetite and sleep disorder in both samples, and the lower the depression level, the less likely the change in behavioral health indicators of appetite and sleep disorder will be. In this way, they can work and live better in the stress event of public health emergencies.

To sum up, this study proposes new ideas for depression, appetite and sleep disorder AST and its influencing mechanism for individuals with Type A personality tendency. Moreover, it is further pointed out that group mental health assistance is necessary, because depression is significantly associated with appetite and sleep disorder in both samples, and has a significant mediating effect on the relationship between Type A personality and appetite and sleep disorder.

Especially among frontline anti-epidemic medical workers, the impact of depression is not only reflected in appetite and sleep disorder, which is a behavioral health indicator, but is also reflected in individuals’ physical health. That is, the higher the level of depression, the higher the likelihood of physical health problems (elevated AST indicator), and thus difficulty of working and living normally in the context of the stressful events of the COVID-19 pandemic. However, in the event of a major public health emergency, the medical system is required to prepare for the tsunami-like treatment needs of the infected population [1]. Therefore, this study calls on all sectors of society to give more attention and assistance to medical staff facing huge industrial challenges.

### 4.3. Multiple Group Comparison

The results of the comparison of multiple groups showed that the relationship between depression, appetite and sleep disorder and the biochemical indicator AST was significantly stronger in the medical staff sample than in the ordinary people sample. However, the relationship between Type A personality and appetite and sleep disorder was significantly stronger in the sample of ordinary people than in the sample of medical personnel.

It follows that, in the context of the global COVID-19 pandemic, it is important for ordinary people with Type A personality traits to disseminate information about health events in order to positively influence their behavioral indicators of appetite and sleep disorder.

At the same time, the results of this study are consistent with those of Bruno et al. (2020), health professionals face specific stressors for a long time, leading to huge risks to their mental health [3]. Additionally, Huang et al. (2020) showed that compared with other occupational groups, medical workers have poorer sleep quality and have a higher risk of mental illness [4].

The results of this study also further confirm the results that the level of depression will further affect the behavioral health indicators of such individuals, appetite and sleep disorder and physiological health indicators AST. Therefore, the results of this study emphasize that it is necessary to pay attention to the mental health status of frontline anti-epidemic medical personnel in the case of major public health emergencies.

### 4.4. Research Significance

#### 4.4.1. Medical Personnel

This study provides a basis for the allocation of medical staff based on personality traits in major public health emergencies. In other words, when selecting individuals with Type A behavioral traits to go to disaster areas with severe epidemic situations to fight against the epidemic on the front line, attention should be paid to the provision of employee assistance (e.g., mental health services). Moreover, attention should be paid to avoiding depression and other psychological problems and to improving their physical health when they go to provide frontline anti-epidemic assistance. For example, the number of Type A medical personnel participating in frontline support activities could be reduced.

In the context of major public health emergencies, a basis for medical staff assistance is proposed. In other words, more attention should be paid to the physical and behavioral health of medical staff with Type A personality who participate in frontline anti-epidemic support activities and medical staff with mental health problems (depression, etc.).

Furthermore, the global COVID-19 outbreak is in its second phase. The results of this study provide a basis on which the WHO, the United Nations, the International Red Cross society, and the Health administration department of the State Council of China can pay attention to the personality traits and psychological distress of health professionals. Furthermore, it is suggested that team leaders should consider the important factor of personality when selecting frontline medical personnel. At the same time, the conclusions of this study should be verified through the classification of medical personnel in different departments. Studies have shown that the level of anxiety and depression in different departments (such as respiratory department and ICU that have close contact with infected patients) in China’s frontline medical staff is different [42].

#### 4.4.2. Ordinary People

At the same time, it provides a basis for the mental health assistance for ordinary people under the COVID-19 pandemic. In the case of major public health emergencies, special attention should be paid to timely psychological crisis intervention for individuals with Type A personality traits.

During the pandemic, specific psychological assistance methods have been described many times in previous studies, such as normalization of strong emotions and stress, satisfaction of basic needs, social support, clear communication and assignment of tasks, flexible working hours and the use of stigmatized psychosocial and psychological help [3].

### 4.5. Limitations and Prospects

Like the majority of studies since 2020, this study focused on the mechanism of the impact of the researchers’ local environment on physical and mental health during the COVID-19 pandemic [4]. As a global preventive work, it lacks international cooperation and a global perspective, and seldom considers the influence of cultural background, a social factor, on the mechanism under study.The variables do not include factors from social psychology, neuroscience or other interdisciplinary fields. It has been shown that the social and neuroscientific implications of COVID-19 must be explored in order to further develop current priorities and long-term strategies for mental health scientific research [43].

## 5. Conclusions

This study identified the relationship between Type A personality traits, depression, appetite and sleep disorder, and biochemical indicators of AST, among ordinary people and frontline anti-epidemic medical staff in the context of COVID-19. The results showed that in both samples, depression was partially mediated between Type A personality and appetite and sleep disorder. In addition, multi-group analysis showed that among the sample of frontline anti-epidemic medical workers, the positive prediction between depression and appetite and sleep disorder and biochemical indicators of AST was stronger. In the sample of ordinary people, the positive prediction between Type A personality and sleep and food status was stronger.

The results of this study may have important implications for psychological assistance of different populations in the context of major public health emergencies. More specifically, our findings suggest that during the COVID-19 pandemic, Type A personality may affect physical and behavioral health in a variety of ways, and the specific process may vary from population to population (i.e., frontline anti-epidemic medical workers and ordinary people).

## Figures and Tables

**Figure 1 ijerph-18-01874-f001:**
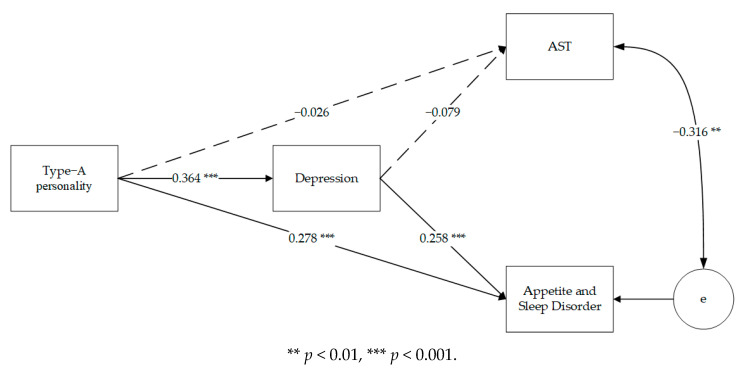
Path model for the group of ordinary people (*n* = 325). Standardized path coefficients among variables are presented. The solid paths are statistically significant.

**Figure 2 ijerph-18-01874-f002:**
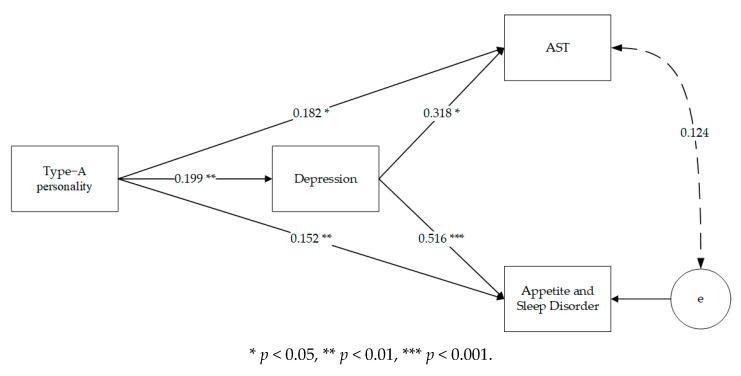
Path model for the frontline anti-epidemic medical staff sample (*n* = 310). Standardized path coefficients among variables are presented. The solid paths are statistically significant.

**Table 1 ijerph-18-01874-t001:** Correlation coefficients and independent samples *t*-test.

Variable	Ordinary People Sample (*n* = 325)	Frontline Anti-Epidemic Medical Personnel Sample (*n* = 310)						
	*M* (*SD*)	*M* (*SD*)	*t*	Cohen’s d	Type A Personality	Depression	Appetite and Sleep Disorder	AST (Aspartate Aminotransferase)
Type A personality	24.446 (7.938)	22.161 (7.312)	3.768 ***	0.299	1	0.199 ***	0.254 ***	0.058
depression	43.246 (10.773)	39.936 (10.285)	3.957 ***	0.314	0.262 **	1	0.546 ***	0.119
appetite and sleep disorder	1.526 (0.547)	1.544 (0.596)	−0.391	−0.031	0.275 ***	0.267 ***	1	0.122
AST	0.156 (0.364)	0.010 (0.995)	5.781 ***	0.211	−0.020	−0.056	−0.091	1
			(df = 259.697)					

** *p* < 0.01, *** *p* < 0.001. The same below. Correlations above the diagonal are for the group of ordinary people; Those below the diagonal are for the group of Frontline anti-epidemic medical personnel.

**Table 2 ijerph-18-01874-t002:** Estimates of indirect effects.

Path	Ordinary People Sample (*n* = 325)			Frontline Anti-Epidemic Medical Personnel Sample (*n* = 310)		
	B(SE)	Lower	Upper	B(SE)	Lower	Upper
Type A personality → depression → AST	−0.029 (0.038)	−0.104	0.046	0.063 (0.037)	−0.010	0.136
Type A personality → depression → appetite and sleep disorder	0.094 ** (0.029)	0.037	0.151	0.103 ** (0.033)	0.038	0.168

** *p* < 0.01.

**Table 3 ijerph-18-01874-t003:** Results of multiple-group model direct effect.

	χ^2^	*df*	Δχ^2^	Δ*df*
Total model (baseline model)	135.918	12	-	-
Model with following constraints				
Type A personality → depression	1.073	1	134.845	11
Type A personality → AST	3.886	3	132.032	9
Type A personality → appetite and sleep disorder	9.689 *	3	126.229	9
depression → AST	4.087 *	1	131.831	11
depression → appetite and sleep disorder	11.413 ***	1	125.505	11

* *p* < 0.05, *** *p* < 0.001.

## Data Availability

The data presented in this study are available on request from the corresponding author. The datasets for this manuscript are not publicly available because the datasets are used only for the team of this article by the permission of the subjects. Requests to access the datasets should be directed to the corresponding author Mei Tian.

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
