# Peer review of "The Pathways from Type A Personality to Physical and Mental Health Amid COVID-19: A Multiple-Group Path Model of Frontline Anti-Epidemic Medical Staff and Ordinary People"

_ijerph, 2021, doi:10.3390/ijerph18041874_

Round 1
Reviewer 1 Report
Here are my minor concerns:
1- In some cases, the paper uses a causal language. Make sure this is nor suggesting any causation.
2- More data are needed on alternative models.
3- More date are needed on validity and reliability and factor structure of the measures used in this study.
4- Instead of bullet points, write the aims as a paragraph.
5- More details are needed on selection of final SEM models.
Other than these comments, I like the paper.
Author Response
Thank you for reminding us of this question, please see the attachment.

Reviewer 2 Report
The topic is current. Please look at these remarks:
1. In Materials and Methods (lines 138-140) "According to the questionnaire response rate... ordinary people (n = 325)". How were these answers collected? Moreover, if these answers were collected by phone or in person, the role of telemedicine in this historic pandemic period needs to be discussed. Add these 2 papers: Montemurro N. Will COVID-19 change neurosurgical clinical practice? Br J Neurosurg. 2020 Jun 1:1-2. doi: 10.1080/02688697.2020.1773399. Ateriya N, Saraf A, Meshram VP, Setia P. Telemedicine and virtual consultation: The Indian perspective. Natl Med J India. 2018 Jul-Aug;31(4):215-218. doi: 10.4103/0970-258X.258220.
2. In Results section (lines 215-218): merge subparagraphs 3.1 and 3.2, please removed "3.1 Independent Samples T-Test and Correlation Analysis", leave "3.1 Path Analysis for Ordinary People and Frontline Anti-epidemic Medical Personnel Samples"
3. "4.5. Limitations and prospects" paragraph (lines 408-418) must be improved. Remove point 1 (lines 408-409), this is not a limitation, you can put this sentence in the discussion section.
4. In Discussion section (lines 283-286) more neurological symptoms (just 2-3 lines) and covid-19 must be discussed.
Author Response

(The authors gave the same response as above.)

Round 2
Reviewer 2 Report
Good.